# FLVCR1a Controls Cellular Cholesterol Levels through the Regulation of Heme Biosynthesis and Tricarboxylic Acid Cycle Flux in Endothelial Cells

**DOI:** 10.3390/biom14020149

**Published:** 2024-01-26

**Authors:** Marta Manco, Giorgia Ammirata, Sara Petrillo, Francesco De Giorgio, Simona Fontana, Chiara Riganti, Paolo Provero, Sharmila Fagoonee, Fiorella Altruda, Emanuela Tolosano

**Affiliations:** 1Molecular Biotechnology Center “Guido Tarone”, Via Nizza 52, 10126 Torino, Italy; marta.manco@kuleuven.be (M.M.); giorgia.ammirata@unito.it (G.A.); sara.petrillo@unito.it (S.P.); francesco.degiorgio@unito.it (F.D.G.); simona.fontana@unito.it (S.F.); chiara.riganti@unito.it (C.R.); sharmila.fagoonee@unito.it (S.F.); fiorella.altruda@unito.it (F.A.); 2Department of Molecular Biotechnology and Health Sciences, University of Torino, Via Nizza 52, 10126 Torino, Italy; 3Laboratory of Tumor Inflammation and Angiogenesis, Center for Cancer Biology, VIB, 3000 Leuven, Belgium; 4Laboratory of Tumor Inflammation and Angiogenesis, Center for Cancer Biology, Department of Oncology, KU Leuven, 3000 Leuven, Belgium; 5Department of Oncology, University of Torino, Via Santena 5/bis, 10126 Torino, Italy; 6Department of Neurosciences “Rita Levi Montalcini”, University of Torino, Corso Massimo D’Azeglio 52, 10126 Torino, Italy; paolo.provero@unito.it; 7Center for Omics Sciences, Ospedale San Raffaele IRCCS, Via Olgettina 60, 20132 Milan, Italy; 8Institute of Biostructure and Bioimaging, CNR c/o Molecular Biotechnology Center “Guido Tarone”, 10126 Torino, Italy

**Keywords:** FLVCR1, FLVCR1a, ALAS1, heme, heme synthesis, cholesterol, endothelial cell

## Abstract

Feline leukemia virus C receptor 1a (FLVCR1a), initially identified as a retroviral receptor and localized on the plasma membrane, has emerged as a crucial regulator of heme homeostasis. Functioning as a positive regulator of δ-aminolevulinic acid synthase 1 (ALAS1), the rate-limiting enzyme in the heme biosynthetic pathway, FLVCR1a influences TCA cycle cataplerosis, thus impacting TCA flux and interconnected metabolic pathways. This study reveals an unexplored link between FLVCR1a, heme synthesis, and cholesterol production in endothelial cells. Using cellular models with manipulated FLVCR1a expression and inducible endothelial-specific *Flvcr1a*-null mice, we demonstrate that FLVCR1a-mediated control of heme synthesis regulates citrate availability for cholesterol synthesis, thereby influencing cellular cholesterol levels. Moreover, alterations in *FLVCR1a* expression affect membrane cholesterol content and fluidity, supporting a role for FLVCR1a in the intricate regulation of processes crucial for vascular development and endothelial function. Our results underscore FLVCR1a as a positive regulator of heme synthesis, emphasizing its integration with metabolic pathways involved in cellular energy metabolism. Furthermore, this study suggests that the dysregulation of heme metabolism may have implications for modulating lipid metabolism. We discuss these findings in the context of FLVCR1a’s potential heme-independent function as a choline importer, introducing additional complexity to the interplay between heme and lipid metabolism.

## 1. Introduction

Feline leukemia virus C receptor 1a (FLVCR1a) is the plasma membrane isoform of the *FLVCR1* gene and was initially identified as the receptor for the retrovirus responsible for cell red aplasia in cats [1]. The FLVCR1a protein exhibits a high degree of conservation across evolutionary scales and is categorized within the SLC49 major facilitator superfamily (MFS) of transporters which specialize in facilitating the transport of small solutes across cellular membranes [2]. Serving as a plasma membrane heme transporter, FLVCR1a exports cytosolic heme into the extracellular environment, hence playing a pivotal role in maintaining cellular heme homeostasis [3]. In particular, a previous study demonstrated a functional connection between FLVCR1a and δ-aminolevulinic acid (ALA) synthase 1 (ALAS1), the rate-limiting enzyme of the heme biosynthetic pathway, that catalyzes the condensation of glycine and succinyl-CoA to form ALA [4]. This reaction occurs in mitochondria, and other than initiating heme synthesis, it contributes to tricarboxylic acid (TCA) cycle cataplerosis by consuming succinyl-CoA [5]. FLVCR1a expression is required to sustain ALAS1 activity, whereas in contrast, FLVCR1a down-modulation results in the inhibition of ALAS1 activity. According to the proposed model, FLVCR1a acts as a positive regulator of heme biosynthesis by exporting excess heme. This export function limits or prevents the negative feedback control of heme on ALAS1, enabling sustained enzymatic activity. Therefore, FLVCR1a plays a pivotal role as a positive regulator in the process of heme biosynthesis [4]. 

The FLVCR1a-ALAS1 axis, in turn, influences the TCA cycle flux, affecting pathways interconnected with it such as fatty acid oxidation, glycolysis, and glutaminolysis [4]. This underscores heme synthesis as a central player in coordinating nutrient utilization. 

This study explores the relationship between the FLVCR1a-ALAS1 axis and cholesterol synthesis, a constitutive process that intersects with heme synthesis at multiple levels. First, cholesterol synthesis also consumes a TCA cycle intermediate, particularly citrate [5]. Moreover, heme is the co-factor of several P450 cytochromes involved in cholesterol synthesis and catabolism. On the other hand, the production of heme A, the prosthetic group of complex IV, depends on isoprenoid farnesyl pyrophosphate coming from the cholesterol biosynthetic pathway. Intriguingly, *FLVCR1a* expression is influenced by cellular cholesterol levels [6], suggesting that FLVCR1a-mediated control of heme synthesis may contribute to cellular cholesterol regulation.

To investigate whether heme synthesis modulates cholesterol production, we focused on endothelial cells (ECs). The proper functioning of ECs relies on active heme synthesis [7,8]. Additionally, the metabolic state of ECs is closely tied to the functionality of FLVCR1a [9,10]. On the other hand, cholesterol, acting as a constituent and organizer of the plasma membrane, serves as a regulator of EC mechanical properties [11,12]. Disruption of cholesterol balance can lead to the impairment of EC functions and ultimately contribute to the development of diseases [13,14]. Therefore, ECs serve as an ideal model to investigate the impact of the FLVCR1a-ALAS1 axis on cholesterol synthesis. Utilizing in vitro and in vivo models with altered FLVCR1a expression, we assessed cholesterol production. Our findings reveal that FLVCR1a, by regulating heme-driven TCA cycle cataplerosis, influences cholesterol production, ultimately impacting the physicochemical properties of the plasma membrane.

## 2. Materials and Methods 

### 2.1. Cell Culture

The human Sk-Hep1 cell line was purchased from the American Type Culture Collection (ATCC, Manassas, VA, USA, catalog n° HTB-52) and was propagated in Dulbecco’s modified Eagle’s medium (DMEM; GIBCO by Thermo Fisher Scientific, Waltham, MA, USA) with 10% FBS (GIBCO), 100 U/mL penicillin, and 100 μg/mL streptomycin (GIBCO). Sk-Hep1 cells were used up to passages 20–25.

Human breast tumor-derived endothelial cells (BTECs) were isolated and characterized in the laboratory of Professor Benedetta Bussolati, Department of Molecular Biotechnology and Health Sciences, University of Torino, Italy [15,16]. BTECs were maintained in EndoGRO-MV-VEGF Complete Culture Media Kit (Merck, Darmstadt, Germany, catalog n° SCME003), supplemented with antibiotics (100 U/mL penicillin and 100 ug/mL streptomycin; Gibco by Thermo Fisher Scientific, catalog n° 15140122). 

Cells were maintained in a 37 °C and 5% CO_2_ air incubator and routinely screened for absence of mycoplasma contamination.

To inhibit heme synthesis, FLVCR1a-overexpressing Sk-Hep1 cells were treated for 24h with 5 mM 5-aminolevulinic acid hydrochloride (Sigma Aldrich, St. Louis, MO, USA, catalog n° A3785). To inhibit citrate export from mitochondria to cytosol, FLVCR1a-silenced Sk-Hep1 cells were treated for 1h with 5 µM iCIC (Sigma Aldrich, catalog n° SML0068).

### 2.2. Gene Silencing and Overexpression

An shRNA against the first exon of human *FLVCR1* gene (Dharmacon, Lafayette, CO, USA, catalog n° RHS4533-NM_014053) was used to specifically downregulate *FLVCR1a* isoform expression, as previously reported [9]. Human FLVCR1a-myc has been previously cloned into pLVX-puro vector [17]. The lentiviruses (i) pLKO.1-sh599 (expressing the shRNA specific for *FLVCR1a*), (ii) pLKO.1-scr (expressing a ‘scramble’ shRNA as control; Dharmacon, catalog n° RHS6848), (iii) pLVX-FLVCR1a-myc and (iv) pLVX-empty as controls were produced in HEK293FT cells. Cells were infected with the lentiviruses in the presence of Sequabrene™ (Sigma Aldrich, catalog n° S2667). Following lentiviral infection, cells were selected with 1.25 μg/mL (Sk-Hep1) or 2 μg/mL (BTEC) puromycin and maintained in 1 μg/mL puromycin (puromycin dihydrochloride from *Streptomyces alboniger*, Sigma Aldrich, catalog n° P8833).

### 2.3. Measurement of Heme Concentration

Intracellular heme concentration was measured using a fluorescence assay, as previously reported [18]. Briefly, Sk-Hep1 or BTECs were collected and 2M oxalic acid was added to them. Samples were heated at 95 °C for 30 min leading to iron removal from heme. Fluorescence (wavelengths: excitation 400 nm–emission 662 nm) of the resultant protoporphyrin was assessed on a GloMax Discover Microplate Reader (Promega Corporation, Madison, WI, USA). The background protoporphyrin content was measured in parallel unheated samples and its value was subtracted. Data were normalized to total protein concentration in each sample. 

### 2.4. Mice

Inducible endothelial-specific *Flvcr1a*-null mice were generated by intercrossing *Flvcr1a^fl/Δ^* mice with *Cdh5-Cre^ERT2^* mice (Tg(Cdh5-Cre/ERT2)1Rha, kindly provided by Ralf H. Adams, on a C57BL/6 background. Mice were genotyped by PCR on genomic DNA from tail biopsies, as previously described [10,19]. To induce *Flvcr1a* gene deletion, 6–8-week-old *Flvcr1a^fl/^^Δ^; Cdh5-Cre^ERT2^* and *Flvcr1a^fl/fl^* mice, hereafter named Flvcr*1a^Δ^*^EC^ and *Flvcr1a^WT^* mice, respectively, were injected intraperitoneally with 75 mg/kg tamoxifen (Sigma Aldrich) for 5 consecutive days. After 3 days from the last injection, mice were sacrificed and ECs (CD45−/CD31+) or liver sinusoidal endothelial cells (LSECs; CD146+) were isolated. For all of the experiments with the inducible knockout model, tamoxifen solution was prepared following The Jackson Laboratory instructions. All of the mice were provided with food and water ad libitum. All experiments with animals were approved by the Italian Ministry of Health.

### 2.5. LSEC Isolation

LSECs were isolated from *Flvcr1a^ΔEC^* and *Flvcr1a^WT^* mice according to Rausa M. et al. with some modifications [20]. Briefly, mice were anesthetized and sacrificed by cervical dislocation. The vena cava was cannulated, and the liver was perfused with pre-warmed liver perfusion medium (Thermo Fisher Scientific, Catalog n° 17701-038) at 37 °C until the liver was completely clear of blood. Then, the liver was perfused with liver digest medium (Thermo Fisher Scientific, Catalog n° 17703-034) at 37 °C until the consistency of the liver changed from an elastic to a deformable state. After perfusion, the liver was removed, teased with scalpels, and incubated for 10 min at 37 °C. Cell suspension was filtered through a 70 μm cell strainer and then centrifuged at 50× *g* for 3 min. The supernatant was centrifugated several times to eliminate hepatocytes. Non-parenchymal cells were pelleted at 400× *g* for 10 min, resuspended in 17.6% Optiprep (STEMCELL Technologies, Cambridge, UK), and stratified onto 8.2% Optiprep. Middle layers, enriched in Kupffer cells and LSECs, were collected, cells were separated by centrifugation, and LSECs were isolated using MACS CD146 MicroBeads (Miltenyi Biotec, Bergisch Gladbach, DE, catalog n° 130-092-007).

### 2.6. EC Isolation 

ECs were isolated from *Flvcr1a^ΔEC^* and *Flvcr1a^WT^* mice as previously described [10]. Briefly, mice lung and liver were dissected and minced into 1–2mm fragments with a scalpel. Tissue pieces were incubated at 37 °C for 60 min in 10 mL of pre-warmed Dulbecco’s phosphate-buffered saline (DPBS) with calcium and magnesium (Lonza Pharma & Biotech, Basel, CH, catalog n BE17-513F) and 2 mg/mL collagenase (collagenase from Clostridium histolyticum, Type I, Sigma Aldrich, catalog n C0130), with regular shacking, until a single cell suspension was obtained. During this incubation, the cells were mechanically dissociated in 10-minute intervals by pipetting. To stop the collagenase activity, DMEM (GIBCO by Thermo Fisher Scientific, catalog n 61965059) containing 10% FBS (GIBCO by Thermo Fisher Scientific, catalog n 10270106) was added to the cell suspension, gently pelleted, and rinsed with PBS. The cells in PBS were then filtered through a 40 μm cell strainer (Corning Life Sciences, Corning, NY, USA, catalog n 352340). Single-cell suspension was centrifuged at 300× *g* for 10 min and ECs were isolated through MACS Technology by using nano-sized MicroBeads, following the manufacturer’s instructions. Particularly, a negative selection was performed using CD45 MicroBeads (Miltenyi Biotec, catalog n 130-052-301). The CD45-negative cell fraction was then pelleted and incubated with CD31 MicroBeads (Miltenyi Biotec, catalog n 130-097-418) to isolate ECs (CD45−/CD31+ cell fraction).

### 2.7. Crystal Violet Assay

Sk-Hep1 cells were plated in 24-well plates and, every 24 h (up to 96 h), were processed. Briefly, the medium was aspired, and cells were gently washed once with PBS. After washing, crystal violet solution was added to each well and the plate was incubated for 30 min at room temperature. Afterward, the plate was washed 5 times with tap water and was left to dry for at least 2 h at room temperature with an open lid until further processing. Before moving to the spectrophotometric measurements, images of plates were captured using a scanner. Afterward, to elute the dye, 10% acetic acid was added in each well and the plate was incubated for 30 min in gentle continuous agitation. The optical density (OD, 560 nm) of each well was measured on a Glomax Multi Detection System (Promega Corporation). Data are expressed as fold increase at 1, 2, and 3 days over day 0 (namely, the first measurement 24 h after plating). Crystal violet assay was performed in technical quadruplicate in three independent experiments.

To monitor the acute toxicity of 5 mM ALA treatment, Sk-Hep1 cells were plated in 24-well plates. After 24 h from plating, one plate was processed (day 0) and the other one was treated with ALA and processed after 24 h (day 1). The optical densities at day 1 were normalized to their respective ones at day 0. Viability data are expressed as percentage (%) over empty cells, which are used as the calibrator. Crystal violet assay was performed in technical quadruplicate in three independent experiments. The same approach was used to monitor the acute toxicity of 5 µM iCIC treatment.

### 2.8. Membrane Fluidity Assay

1 × 10^6^ BTECs or Sk-Hep1 cells were stained with the membrane fluidity kit (Marker Gene Technologies, Eugene, OR, USA, catalog n° M0271), according to the manufacturer’s protocol, exploiting the properties of the membrane-bound fluorophore of the kit that shifts from monomer in a more rigid membrane (with a maximal emission wavelength of 372 nm, Ii) to excimer formation in a more fluid membrane (with a maximal emission wavelength of 470 nm, Ie). Results were expressed as Ie/Ii ratio, the index of membrane fluidity. Membrane fluidity assay was performed in technical triplicate in three independent experiments.

### 2.9. Membrane Cholesterol Content

Sk-Hep1 cells, BTECs, or LSECs were lysed in 0.5 mL of lysis buffer (HEPES, pH 7.4, 20 mM, KCl 10 mM, MgCl_2_ 2 mM, EGTA 1, and DTT 1 mM) by gentle scraping. Following an incubation time of 15 min on ice, the lysate was passed through a 25-gauge needle 10 times and kept on ice for a further 20 min. To remove nuclei and mitochondria, samples were centrifuged at 10,000× *g* for 5 min at 4 °C. The supernatant was subsequently ultra-centrifuged (Beckman Coulter’s, rotor 70Ti) at 100,000× *g* for 1 h at 4 °C. The new supernatant (i.e., the cytoplasm fraction) was discarded, whereas the pellet (i.e., the membrane fraction) was rinsed with 200 µL of PBS and kept at −80 °C until further processing. The protein content of the membrane extracts was measured on a 50 µL aliquot, using the BCA Protein Assay kit (Sigma Aldrich). The amount of cholesterol was measured on 100 µg membrane protein using the fluorimetric Cholesterol/Cholesteryl Ester Assay Kit—Quantitation (Abcam, Cambridge, UK), as per the manufacturer’s instructions. The results are expressed as µmol cholesterol/mg membrane protein, based on the calibration curve. Membrane cholesterol assay was performed in three independent experiments.

### 2.10. Filipin Staining

BTECs were fixed with 4% paraformaldehyde for 15 min and then incubated with 1.5 mg/mL glycine for 10 min. After that, the cells were incubated in 50 μg/mL (PBS diluted) filipin III ready-made solution (Merck, catalog n° SAE0087) for 2 h at room temperature [21]. Images were acquired by a NIKON-VICO widefield microscope using a 405 nm excitation wavelength. Total cellular filipin signal was analyzed by using ImageJ software Version 1.54h.

### 2.11. Total (Intracellular) Cholesterol

The amount of cholesterol was measured in 100 µg total protein by using the fluorimetric Cholesterol/Cholesteryl Ester Assay Kit–Quantitation (Abcam), according to the manufacturer’s instructions. The results are expressed as µmoles cholesterol/mg cell protein, according to the calibration curve.

### 2.12. De Novo Synthesis of Farnesyl Pyrophosphate (FPP) and Geranylgeranyl Pyrophosphate (GGPP)

The FPP and GGPP synthesis rate was evaluated by incubating 1 × 10^5^ Sk-Hep1 cells or BTECs with the radiolabeled FPP/GGPP precursor [3H]-acetate (Amersham Bioscience, Little Chalfont, Buckinghamshire, UK) for 24 h, as previously reported [22]. FPP and GPP synthesis was measured by liquid scintillation and expressed as fmoles [3H]FPP or [3H]GGPP/10^6^ cells, according to relative calibration curves.

### 2.13. De Novo Synthesis of Cholesterol 

Cholesterol synthesis rate was evaluated by incubating cells with a radiolabeled cholesterol precursor, as previously reported [23]. Briefly, Sk-Hep1 cells or BTECs were plated in 6-well plates and, the following day, were incubated for 24 h with 1 µCi/mL [^3^H]-acetate (Amersham Bioscience). After two washes with PBS, cells were collected and transferred into glass microcentrifuge tubes. Lipids were then extracted in a 1.5 mL solution of methanol/hexane (1:2, *v*/*v*) and subjected to 1 h of shaking at room temperature. The upper phase was transferred to a new tube, while the lower phase was supplemented with 1 mL hexane and stirred overnight. The new upper phase was added to the previous one and the solvent was evaporated for 24 h. The lipids contained in the pellet were resuspended into 20 µL chloroform and separated by thin layer chromatography (TLC; pre-coated LK6DWhatman silica gels, Merck) using a 1:1 (*v*/*v*) ether/hexane solution as the mobile phase for 30 min. Standards of 10 µg/mL cholesterol were run in parallel. The silica gel plates were exposed for 1 h to an iodine-saturated atmosphere, and the migrated spots corresponding to cholesterol were cut out. Their radioactivity was measured by liquid scintillation, using a Tri-Carb Liquid Scintillation Analyzer (PerkinElmer, Waltham, MA, USA). Cholesterol was expressed as relative change in fmoles [^3^H]-cholesterol/10^6^ cells, according to the titration curve. The assay was performed in technical triplicate in three independent experiments.

### 2.14. RNA Extraction and Real-Time PCR Analysis

Total RNA was extracted using TRIzol^TM^ reagent, according to the manufacturer’s instructions (Thermo Fisher Scientific). 0.5-1 μg total RNA was retro-transcribed into complementary DNA (cDNA) using a High-Capacity cDNA Reverse Transcription Kit (Thermo Fisher Scientific). Quantitative real-time PCR (qRT-PCR) was carried out using Platinum^TM^ Quantitative PCR SuperMix-UDG w/ROX (Thermo Fisher Scientific) and was performed on the 7900HT Fast Real-Time PCR System (Thermo Fisher Scientific) or on the QuantStudio 6 Flex Real-Time PCR System (Thermo Fisher Scientific) with 96-well or 384-well plates, respectively. Specific primers and probes were designed using Primer Express Software Version 3.0 (Thermo Fisher Scientific). Analysis was performed using the 2^–∆∆Ct^ method. Relative transcript abundance, normalized to 18S mRNA expression, is expressed as a fold increase over the calibrator sample(s).

### 2.15. Western Blot

Cells were lysed in RIPA buffer supplemented with 1 mM phosphatase inhibitor cocktail (Sigma Aldrich), 1 mM PMSF (Sigma Aldrich), and a protease inhibitor cocktail (Roche, Basel, CH), and protein concentration was determined by Bradford assay. Protein extract (10 µg) was incubated for 10 min at 37 °C with 1 µL of PNGase-F (Sigma Aldrich) to remove protein glycosylation. A loading buffer (supplemented with 8% 2-mercaptoethanol) was added and samples were incubated for 5 min at 37 °C. The primary antibodies are as follows: mouse monoclonal anti-FLVCR1 (C-4) (Santa Cruz Biotechnology, Dallas, TX USA, catalog n° sc-390100; 1:500); mouse monoclonal anti-Vinculin (Sigma Aldrich, catalog n° SAB4200080, 1:8000). Immunoreactions were visualized by chemiluminescence with the ChemiDoc Imaging System (Bio-Rad, Hercules, CA USA).

### 2.16. Mitochondrial Extraction and ALAS Activity

Mitochondria were extracted as reported in [24]. Cells were lysed in 0.5 ml mitochondria lysis buffer (50 mM Tris-HCl, 100 mM KCl, 5 mM Mg Cl_2_, 1.8 mM ATP, and 1 mM EDTA at pH7.2), supplemented with protease inhibitor cocktail III (Sigma Aldrich), 1 mM phenylmethylsulfonyl fluoride (PMSF), and 250 mM NaF. Samples were clarified by centrifugation at 650× *g* for 3 min at 4 °C. Supernatants were collected and centrifuged at 13,000× *g* for 5 min at 4 °C. Pellets, containing mitochondria, were washed once with lysis buffer and resuspended in 0.25 mL mitochondria resuspension buffer (250 mM sucrose, 15 mM K_2_HPO_4_, 2 mM MgCl_2_, and 0.5 mM EDTA). Part of the mitochondrial sample was sonicated and used to measure the mitochondrial proteins with the BCA Protein Assay kit (Sigma Aldrich) and for quality control: 10 µg of each sonicated sample was analyzed by SDS-PAGE and immunoblotting with an anti-porin antibody (Abcam; clone 20B12AF2) to confirm the presence of mitochondrial proteins in the extracts. The remaining 200 µL was used for ALAS activity measurement, according to [25]. A total of 10 µL of the reaction’s product was injected into the Waters Acquity ultra-performance liquid chromatography **(**UPLC) system, equipped with a binary solvent manager, sample manager, photodiode array detector (PDA), fluorescence detector, column heater, and an Acquity UPLC BEH C18, 1.7 μM, 2.1 × 100 mm column. Detection of ALA-derivative was performed according to [25], setting the detector with λ excitation of 370 nm and λ emission of 460 nm, and the range of the PDA scanner was set between 210 and 500 nm. The results were converted into nmol/min based on a calibration curve and expressed as nmol/min/mg mitochondrial proteins.

### 2.17. C-Isotope Labeling Experiments

Sk-Hep1 cells were seeded in 6-well plates, and after 24 h, the medium was replaced with 10% FBS-supplemented DMEM (DMEM, no glucose and no glutamine; GIBCO by Thermo Fisher Scientific) containing 4 mM unlabeled glutamine and 25 mM ^13^C_6_-glucose for glucose-tracing experiments; or with 10% FBS-supplemented DMEM (DMEM, no glucose and no glutamine; GIBCO by Thermo Fisher Scientific) containing 25 mM unlabeled glucose and 4 mM ^13^C_5_-glutamine for glutamine tracing experiments. The labeled medium was maintained for 24 h, and at the end of the incubation, cells were rapidly washed three times with ice-cold 0.9% NaCl solution and metabolites were extracted following the ‘sample preparation and delivery protocol manual’ received from the VIB Metabolomics Core (MEC, Leuven Belgium). Briefly, cells were covered with extraction buffer (stored at −80 degrees, buffer provided by MEC), and after 2–3 min on ice, cells were scraped. The extracts were centrifuged at 20,000× *g* for 15 min at 4 °C. Protein pellets were used to measure the protein content using the BCA Protein Assay kit (Sigma Aldrich) and supernatants were collected into new tubes and provided to MEC for liquid chromatography–mass spectrometry (LC-MS).

### 2.18. Bioinformatic Analyses

The publicly available microarray dataset GSE47067 from [26] has been used to investigate the expression of *Flvcr1* in ECs derived from several murine tissues. *Flvcr1* is represented by two probes on the microarray (#10361065 and #10361075), and the expression of the two probes in the different tissue-specific ECs is represented in the plots. Following normalization, overall median gene expression is zero, so that expression levels above 0 can approximately be considered to indicate detection. The publicly available RNA-seq dataset GSE164878 from [27] on the interactive website www.shiny.lvbrg.barcelona/lsec (accessed in 18 December 2023) has been used to investigate the expression of heme synthesis- and cholesterol synthesis-related genes in cirrhotic versus healthy rat LSECs. Log_2_ fold change and FDR (false discovery rate) have been downloaded.

### 2.19. Statistical Analyses 

Statistical analyses were conducted in GraphPad Prism v5.0 and v7.0 (GraphPad Software, Inc., La Jolla, CA, USA, https://www.graphpad.com; RRID:SCR_002798). Results are expressed as mean ± SEM. Statistical analyses were performed by employing Student’s *t*-test or multiple unpaired *t*-tests, and one-way ANOVA when two or more experimental conditions were compared, respectively. A *p*-value of <0.05 was considered statistically significant.

## 3. Results

### 3.1. FLVCR1a Is a Positive Regulator of Heme Synthesis in Endothelial Cells

We have previously established the functional interplay of FLVCR1a with ALAS1, emphasizing its necessity for sustaining heme synthesis [4]. To ascertain the universality of this relationship in ECs, we manipulated *FLVCR1a* expression, creating both loss-of-function and gain-of-function models in these cells. We then assessed heme synthesis and content in these models. Our investigation centered on Sk-Hep1 cells, a well-recognized model for liver sinusoidal endothelial cells (LSECs) [28] and on breast-tumor-derived endothelial cells (BTECs), representing the activated endothelium [16] (Figure 1A,B). *FLVCR1a* overexpression as well as *FLVCR1a* silencing in Sk-Hep1 cells had no impact on cell viability (Appendix A,) while *FLVCR1a* down-modulation in BTECs resulted in a slight but significant reduction in cell viability [10].

We inferred heme synthesis by measuring ALA production in mitochondrial extracts obtained from *FLVCR1a*-silenced and *FLVCR1a*-overexpressing cells. As expected, ALAS1 activity was significantly lower and higher in *FLVCR1a*-silenced and *FLVCR1a*-overexpressing cells, respectively, compared to their controls (Figure 1C,D). In Sk-Hep1 cells, the modulation of ALAS1 activity did not result in altered intracellular heme content (Figure 1E). On the contrary, in BTECs, *FLVCR1a* silencing or overexpression resulted in a slight but significant increase or decrease in heme content, respectively (Figure 1F), supporting the notion that *FLVCR1a* expression controls intracellular heme accumulation.

### 3.2. FLVCR1a Expression Regulates Cholesterol Production in Endothelial Cells

Previous studies have demonstrated the regulatory influence of the ALAS1-FLVCR1a axis on the flux of the TCA cycle and processes governed by the TCA cycle [4,29]. Our objective was to investigate whether FLVCR1a, through its control of ALAS1, can impact on cholesterol synthesis. This biochemical pathway begins with the condensation of two acetyl-CoA molecules catalyzed by ATP citrate lyase, which utilizes citrate derived from the TCA cycle as a substrate.

To assess cholesterol production, we measured intracellular radiolabeled cholesterol after incubation of Sk-Hep1 cells or BTECs with a radiolabeled cholesterol precursor (i.e., [^3^H]-acetate) for 24 h. FLVCR1a-silenced and FLVCR1a-overexpressing cells exhibited high and low cholesterol synthesis, respectively, compared to their controls (Figure 2A). Consistently, the production of FPP, an intermediate of the pathway, was increased or reduced in FLVCR1a-silenced or FLVCR1a-overexpressing cells, respectively (Figure 2B). The same result was obtained with the production of GGPP that derives from FPP (Figure 2C).

To confirm and implement these data, we performed untargeted metabolomic experiments by incubating Sk-Hep1 cells with either ^13^C fully labeled glucose or glutamine and by measuring, after 24 h, the incorporation of these isotopes into cholesterol (Figure 2D). As previously reported [30,31], we observed that glucose was the major carbon source for cholesterol synthesis. In agreement with the data shown in Figure 2A, FLVCR1a-silenced Sk-Hep1 cells produced more cholesterol than controls, whereas in contrast, FLVCR1a-overexpressing cells showed reduced cholesterol production (Figure 2E). These data suggest that heme-driven cataplerosis of the TCA cycle might regulate the amount of citrate available for cholesterol synthesis, i.e., when more succinyl-CoA is used by ALAS1, less citrate is available for the mevalonate pathway and vice versa. Consistently, oxidation of either glucose or glutamine through the TCA cycle did not change upon FLVCR1a modulation (Figure 2F,G). On the contrary, reductive carboxylation of glutamine, a pathway usually potentiated to sustain de novo lipogenesis [30], was increased or reduced in FLVCR1a-silenced or FLVCR1a-overexpressing Sk-Hep1 cells, respectively (Figure 2H), further strengthening the concept that when heme synthesis is inhibited, more citrate is released from mitochondria for cholesterol synthesis and vice versa.

To definitively prove that heme-driven cataplerosis of the TCA cycle controls citrate availability for cholesterol synthesis, we performed two rescue experiments (Appendix A). On the one hand, FLVCR1a-silenced cells were treated with an inhibitor of the mitochondrial citrate carrier (hereinafter referred to as iCIC) to limit citrate efflux from mitochondria to cytosol where cholesterol synthesis occurs [32]. On the other hand, FLVCR1a-overexpressing cells were treated with ALA, which is converted into protoporphyrin IX (PPIX) and heme, thus inhibiting ALAS1 through a negative feedback loop (Appendix A). First, we ascertained that the treatment with iCIC or ALA did not affect the viability of FLVCR1a-silenced or FLVCR1a-overexpressing cells, respectively (Appendix A). Subsequently, we observed that the treatment of FLVCR1a-silenced cells with iCIC, namely, the inhibition of citrate efflux, restored cholesterol synthesis at a level comparable to that of untreated control cells (Figure 2I). On the other hand, the treatment of FLVCR1a-overexpressing cells with ALA, namely, the inhibition of ALAS1-mediated succinyl-CoA utilization, was sufficient to rescue cholesterol synthesis in FLVCR1a-overexpressing cells (Figure 2J). 

These results indicate that citrate efflux and cholesterol synthesis strictly depend on FLVCR1a-controlled ALAS1 activity and succinyl-CoA consumption. 

### 3.3. FLVCR1a Modulates Cellular Cholesterol Levels

The data presented in the preceding section indicates a potential role of FLVCR1a in regulating intracellular cholesterol levels. This was substantiated through the quantification of cellular cholesterol levels using the cholesterol-binding dye filipin (Figure 3A). Given that cholesterol is a crucial constituent of the plasma membrane, influencing membrane tension [33], we hypothesized that altering FLVCR1a expression, either through silencing or overexpression, could impact membrane cholesterol content and membrane fluidity. In line with this hypothesis, we observed that cells with silenced FLVCR1a and those with overexpressed FLVCR1a displayed elevated and reduced levels of cholesterol in the membrane, respectively, compared to their respective controls (Figure 3B). Subsequently, we measured membrane fluidity in both FLVCR1a-silenced and FLVCR1a-overexpressing cells, as well as in control cells. The first one exhibited a more rigid membrane, whereas the second one displayed a more fluid membrane, compared to their respective controls (Figure 3C). In agreement with the data shown in Figure 2I,J, the treatment of FLVCR1-silenced cells with iCIC and of FLVCR1a-overexpressing cells with ALA was able to restore membrane cholesterol content and membrane fluidity at a level comparable to that of their respective untreated controls (Figure 3D,E), thus strengthening the functional link between heme synthesis and cellular cholesterol level.

### 3.4. FLVCR1a Modulates Cellular Cholesterol Levels In Vivo

To validate the relevance of our findings, we initially examined the publicly accessible microarray gene expression dataset GSE4706716, specifically focusing on *Flvcr1* expression in microvascular ECs. Two probe sets targeting *Flvcr1* (#10361065 and #10361075) were identified. *Flvcr1* was found to be expressed in various murine microvascular ECs, with the highest expression observed in the liver (Figure 4A). This finding was corroborated by qRT-PCR analysis conducted on ECs isolated from the liver and lung of wild-type mice (Figure 4B). Remarkably, analysis of publicly available RNA-seq data [27] on LSECs derived from normal rats and three distinct rat models of liver damage (i.e., CCl_4_, carbon tetrachloride model; TAA, thioacetamide model; BDL, bile duct ligation model [34,35]) revealed a significant downregulation and upregulation, respectively, of the expression of genes related to heme synthesis and cholesterol synthesis following injury (Figure 4C). This supports the existence of a functional connection between heme synthesis and cholesterol production.

To further investigate this point, we took advantage of inducible endothelial-specific *Flvcr1a*-deficient mice (*Flvcr1a^ΔEC^*) and evaluated cellular cholesterol levels in ECs isolated from the liver (both the CD45−/CD31+ endothelial component and the sinusoidal specific CD146+ cell component) and the lung (CD45−/CD31+) of these mice compared to those obtained from wild-type controls (Figure 4D). As expected, ALAS activity was reduced in ECs isolated from *Flvcr1a^ΔEC^*mice compared with those from *Flvcr1a^WT^* mice (Figure 4E), confirming that also in vivo, FLVCR1a is a positive regulator of heme synthesis. Consistently with in vitro data, reduced heme synthesis in FLVCR1a-deficient ECs resulted in increased intracellular cholesterol content and a higher cholesterol level in membranes (Figure 4F,G).

Collectively, our data demonstrate that FLVCR1a expression in ECs regulates the intracellular cholesterol level by controlling heme synthesis.

## 4. Discussion 

Here, we highlight a hitherto undiscovered relationship between FLVCR1a and cholesterol production controlled by FLVCR1a’s influence on heme-driven TCA cycle cataplerosis. The TCA cycle, a central hub for cell energetic metabolism, is considered an amphibolic pathway as, on the one hand, it converts nutrients into energy, and on the other hand, it produces intermediates that are used in several anabolic reactions to produce lipids, amino acids, and nucleotides. Given this dual role in both maintaining energy production and in provisioning key anabolic substrates, TCA cycle activity is tightly regulated [5]. In general, the utilization of any TCA cycle intermediate, a process termed cataplerosis, requires compensatory input to sustain TCA cycle activity. Heme synthesis, by consuming succinyl-CoA, is a cataplerotic reaction of the TCA cycle. At the same time, cholesterol synthesis may also be considered a process that consumes TCA intermediates as it starts in the cytosol from acetyl-CoA produced by ATP-citrate lyase using mitochondria-derived citrate as a substrate [5]. Our data demonstrate that the amount of succinyl-CoA and citrate in mitochondria are balanced to maintain TCA cycle flux and that heme synthesis, controlled by FLVCR1a, regulates citrate availability for cholesterol synthesis. Thus, when FLVCR1a is overexpressed and heme synthesis is stimulated, less citrate is available for cholesterol production and vice versa (Figure 5). 

The regulation of heme synthesis is primarily governed by heme itself through a negative feedback mechanism which regulates the expression and activity of ALAS1 [36]. Based on the earlier proposition of FLVCR1a as a heme exporter [3], we previously proposed that this transporter controls a pool of intracellular regulatory heme that in turn regulates ALAS1 [4,37]. While this model aligns with the current findings, recent discoveries indicate the possibility of alternative explanations. Specifically, FLVCR1a has been shown to function as a choline importer, playing a crucial role in phosphatidylcholine production—the primary component of cell membranes [38,39,40]. Consequently, the positive influence of FLVCR1a on heme synthesis might not be directly orchestrated by heme control but rather indirectly through choline metabolism. This aspect remains an unresolved question for future investigations.

Interestingly, hydroxy-3-methylglutaryl-CoA synthase 1 (HMGCS1), a pivotal enzyme in the mevalonate pathway responsible for cholesterol synthesis, has been recognized as an essential gene for proliferation in FLVCR1a knockout HEK293T cells [38]. Our data offer a plausible explanation for this essentiality by suggesting that an elevated production of cholesterol is necessary to offset the block of heme-driven cataplerosis. This compensation becomes crucial to uphold the TCA cycle flux, a crucial factor for sustaining cellular proliferation.

Furthermore, the notion that heme-driven cataplerosis serves as a mechanism to regulate TCA cycle flux aligns with prior research. This includes the discovery that the heme oxygenase 1 (HMOX1) gene, responsible for heme degradation, exhibits synthetic lethality in fumarate hydratase (FH) mutated cells [41]. In these cells, heightened heme synthesis compensates for the impaired TCA cycle flux resulting from FH deficiency.

Beyond its role in cholesterol production, the regulation of TCA cycle cataplerosis by FLVCR1a is anticipated to influence various pathways intricately connected with the TCA cycle. According to this view, heme synthesis is not merely a process required for hemoprotein production but is part of a network of strictly interlinked metabolic pathways that control cell fitness and allow for cell adaptation in response to internal and external stimuli. This concept is in line with previous observations that heme synthesis is a critical determinant of oxidative metabolism of cancer cells and represents a metabolic dependency for pancreatic and lung tumors [4,42,43].

In the present study, we have shown that the modulation of cholesterol production driven by FLVCR1a in ECs is biologically relevant both in vitro and in vivo as it regulates membrane cholesterol content and membrane fluidity. Maintaining cholesterol homeostasis is essential for supporting proper vascular development and sustaining endothelial functions [44,45,46]. Interestingly, FLVCR1a is required by ECs to accomplish proper angiogenesis as endothelial-specific *Flvcr1a*-null mice show defective developmental and adult angiogenesis characterized by reduced cell proliferation and impaired ability to form a regular vascular network [9,10]. Our data suggest that alterations in cholesterol production and its subsequent accumulation in membranes due to the inhibition of heme synthesis could contribute to these abnormalities. Indeed, cholesterol abundance and tethering at the plasma membrane modulates cholesterol-enriched lipid microdomains, namely, lipid rafts, which in turn are crucial components required for proper angiogenic signaling, particularly in pathways involving vascular endothelial growth factor receptor 2 (VEGFR2) and Notch. Furthermore, changes in membrane fluidity driven by cholesterol could potentially affect the migratory phenotype of ECs [44,47], thereby contributing to angiogenic defects associated with FLVCR1a.

In addition, in LSECs, membrane tension is a crucial determinant in the formation/maintenance of fenestrae, the hallmark of this highly specialized type of endothelium. It has been proposed that fenestrae originate from a process of plasma membrane invagination and fusion with the basolateral membrane. This process is governed by the physical–chemical properties of the plasma membrane. In particular, reduced membrane tension leads to membrane invaginations, whereas increased membrane tension hampers these invaginations [48,49]. Cholesterol plays a key role in regulating membrane tension and fenestrae formation [50]. Indeed, cholesterol depletion and cholesterol accumulation in the plasma membrane lead to an increase and decrease in the number of fenestrations, respectively [48,49]. Hence, it is plausible to hypothesize that FLVCR1a, by modulating membrane cholesterol content, can participate in the membrane dynamics necessary for fenestrae formation and maintenance. Additionally, it is imperative to consider the potential heme-independent function of FLVCR1a as a choline importer, which could further contribute to the regulation of membrane composition and dynamics.

Finally, the FLVCR1 gene is overexpressed in most tumors, and this contributes to metabolic adaptation that sustains tumor growth [4,51]. If the inverse correlation between FLVCR1a and cholesterol production also occurs in cancer cells, the sustained heme synthesis promoted by FLVCR1a overexpression might contribute to membrane fluidity that is required for cell growth and migration. According to this hypothesis, FLVCR1a-silenced cancer cells show increased hydroxy-methyl glutaryl CoA (HMG-Coa), an intermediate of cholesterol synthesis [4]. 

In summary, our findings demonstrate that FLVCR1 influences cellular cholesterol levels through the regulation of heme biosynthesis and TCA cycle flux, underscoring the interconnected nature of heme metabolism with various metabolic pathways. The recent discovery of FLVCR1a functioning as a choline importer introduces an additional layer of complexity. This aspect deserves future exploration in order to elucidate the intricate interplay between heme, choline, and lipid metabolism.

## 5. Declaration of Generative AI and AI-Assisted Technologies in the Writing Process

During the preparation of this work, the authors used ChatGPT in order to improve language and readability. After using this tool, the authors reviewed and edited the content as needed and take full responsibility for the content of the publication.

## Figures and Tables

**Figure 1 biomolecules-14-00149-f001:**
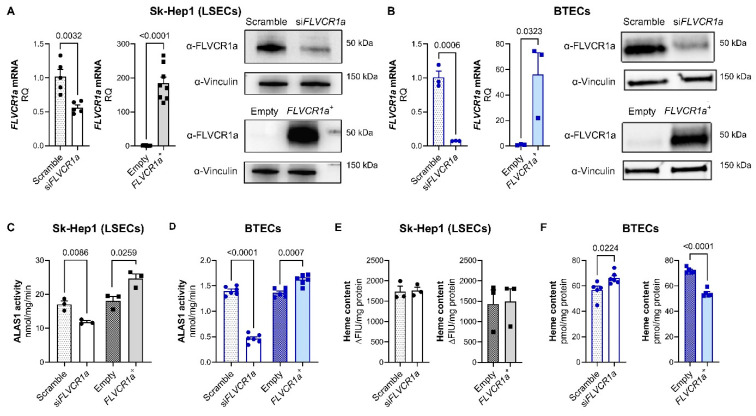
FLVCR1a is a positive regulator of heme synthesis in endothelial cells. (**A**) Generation of FLVCR1a loss- and gain-of-function models. (**A**; left panel) mRNA expression levels of *FLVCR1a* in FLVCR1a-silenced (*siFLVCR1a*) and -overexpressing (*FLVCR1a^+^*) Sk-Hep1 cells. Normalization was achieved using 18S as housekeeping gene. Data are expressed as mean ± SEM of relative quantification using the 2^–∆∆Ct^ method over their respective controls (*n* = 5–8). Unpaired *t*-test was performed. (**A**; right panel) Representative images of Western blot showing FLVCR1a (long exposure, top; short exposure, bottom) and vinculin (housekeeping) expression in FLVCR1a-modulated Sk-Hep1 cells. (**B**) Generation of FLVCR1a loss- and gain-of-function models. (**B**; left panel) mRNA expression levels of *FLVCR1a* in FLVCR1a-silenced (*siFLVCR1a*) and -overexpressing (*FLVCR1a^+^*) BTECs. Normalization was achieved using 18S as housekeeping gene. Data are expressed as mean ± SEM of relative quantification using the 2^–∆∆Ct^ method over their respective controls (*n* = 3). Unpaired *t*-test was performed. (**B**; right panel) Representative images of Western blot showing FLVCR1a (long exposure, top; short exposure, bottom) and vinculin (housekeeping) expression in FLVCR1a-modulated BTECs. (**C**) ALAS activity in mitochondria of FLVCR1a-modulated Sk-Hep1 cells is expressed as nmol/min/mg protein. Data are expressed as mean ± SEM (*n* = 3). Unpaired *t*-test was performed. (**D**) ALAS activity in mitochondria of FLVCR1a-modulated BTECs is expressed as nmol/min/mg protein. Data are expressed as mean ± SEM (*n* = 6). Unpaired *t*-test was performed. (**E**) Intracellular heme content of FLVCR1a-modulated Sk-Hep1 cells is expressed as fluorescence (ΔFIU)/mg protein. Data are expressed as mean ± SEM (*n* = 3). Unpaired *t*-test was performed. (**F**) Intracellular heme content of FLVCR1a-modulated BTECs is expressed as pmol/mg protein. Data are expressed as mean ± SEM (*n* = 5–6). Unpaired *t*-test was performed. si*FLVCR1a*: FLVCR1a-silenced cells; scramble: respective control; *FLVCR1a^+^*: FLVCR1a-overexpressing cells; empty: respective control; LSECs: liver sinusoidal endothelial cells; BTECs: breast tumor-derived endothelial cells. Original images that can be found at Appendix A.

**Figure 2 biomolecules-14-00149-f002:**
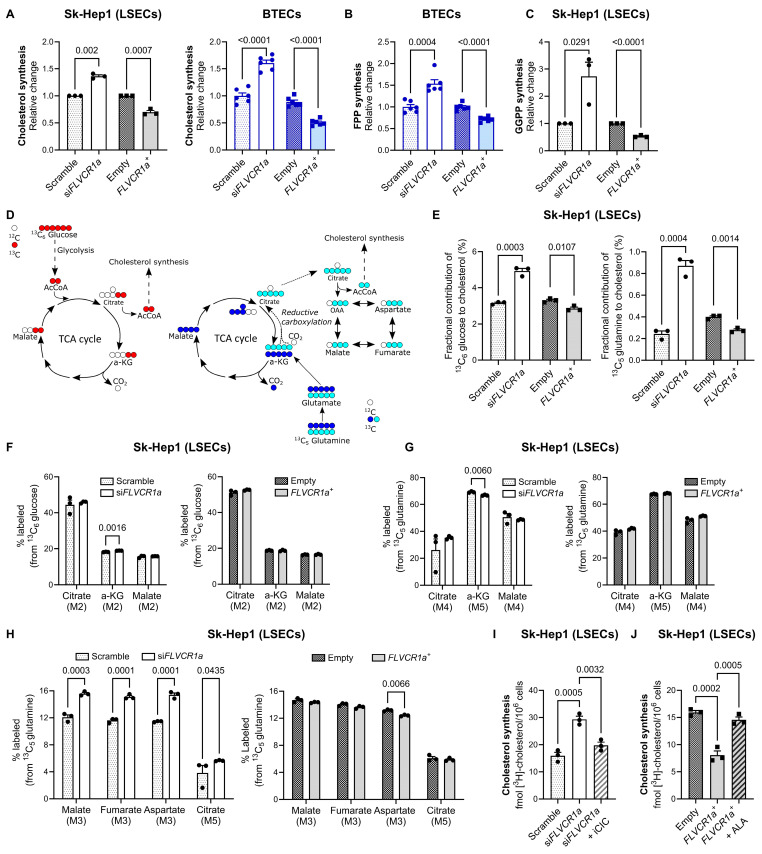
FLVCR1a expression regulates cholesterol production in endothelial cells. (**A**) De novo cholesterol synthesis rate is expressed as relative change versus control cells (scramble or empty). Data are expressed as mean ± SEM (*n* = 3, three independent experiments, left panel; *n* = 6, six independent experiments, right panel). Unpaired *t*-test was performed. (**B**) De novo FPP synthesis rate is expressed as relative change versus control cells (scramble or empty). Data are expressed as mean ± SEM (*n* = 6, six independent experiments). Unpaired *t*-test was performed. FPP, farnesyl pyrophosphate. (**C**) De novo GGPP synthesis rate is expressed as relative change versus control cells (scramble or empty). Data are expressed as mean ± SEM (*n* = 3, three independent experiments). Unpaired *t*-test was performed. GGPP, geranylgeranyl pyrophosphate. (**D**) Schematic representation of fully labeled (^13^C_6_) glucose (depicted in red) and (^13^C_5_) glutamine (depicted in dark blue for oxidation and in light blue for reductive carboxylation) metabolic flux. TCA, tricarboxylic acid; AcCoA, acetyl-CoA; OAA, oxoalacetate; α-KG, α-ketoglutarate. (**E**) Fractional contribution of glucose (left panel) and glutamine (right panel) to cholesterol. Data are expressed as mean ± SEM (*n* = 3). Multiple unpaired *t*-tests were performed. (**F**) Fractional contribution of glucose to TCA intermediates in FLVCR1a-silenced (left panel) and FLVCR1a-oxerexpressing (right panel) Sk-Hep1 cells, compared to their respective controls. Data are expressed as mean ± SEM (*n* = 3). Multiple unpaired *t*-tests were performed. (**G**) Fractional contribution of glutamine to TCA intermediates in FLVCR1a-silenced (left panel) and FLVCR1a-oxerexpressing (right panel) Sk-Hep1 cells, compared to their respective controls. Data are expressed as mean ± SEM (*n* = 3). Multiple unpaired *t*-tests were performed. (**H**) Reductive carboxylation in FLVCR1a-modulated cells. Relative abundance of reductive carboxylation-specific mass isotopomers in FLVCR1a-silenced (left panel) and FLVCR1a-oxerexpressing (right panel) Sk-Hep1 cells, compared to their respective controls. Data are expressed as mean ± SEM (*n* = 3). Multiple unpaired *t*-tests were performed. (**I**) Rescue experiments in FLVCR1a-silenced Sk-Hep1 cells. De novo cholesterol synthesis rate is expressed as fmol of radiolabeled cholesterol over 1 × 10^6^ cells. Data are expressed as mean ± SEM (*n* = 3, three independent experiments). Unpaired *t*-test was performed. (**J**) Rescue experiments in FLVCR1a-overexpressing Sk-Hep1 cells. De novo cholesterol synthesis rate is expressed as fmol of radiolabeled cholesterol over 1 × 10^6^ cells. Data are expressed as mean ± SEM (*n* = 3, three independent experiments). Unpaired *t*-test was performed. si*FLVCR1a*: FLVCR1a-silenced cells; scramble: respective control; *FLVCR1a^+^*: FLVCR1a-overexpressing cells; empty: respective control; iCIC: mitochondrial citrate carrier inhibitor; ALA: δ-aminolevulinic acid (heme synthesis inhibitor); LSECs: liver sinusoidal endothelial cells; BTECs: breast tumor-derived endothelial cells.

**Figure 3 biomolecules-14-00149-f003:**
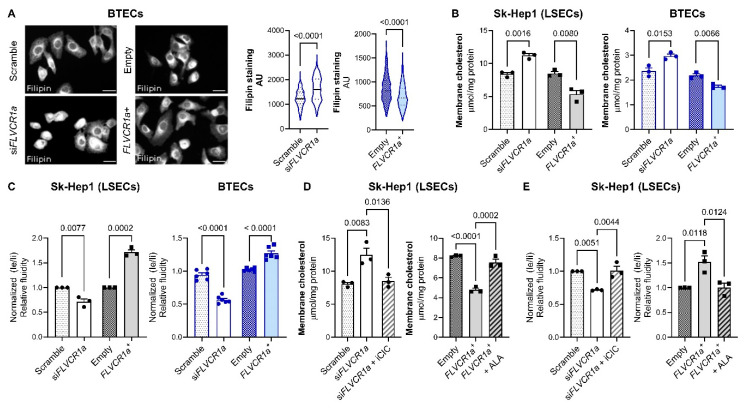
FLVCR1a modulates cellular cholesterol levels. (**A**) Representative images (left panel) and relative quantification (right panel) of filipin staining in BTECs. Data are expressed as A.U. (arbitrary unit) indicating signal intensity normalized over the cell area. (**B**) FLVCR1a level perturbation affects physical–chemical properties of plasma membrane. Cholesterol content in plasma membrane is expressed as µmol/mg protein in both Sk-Hep1 cells and BTECs (left and right panel, respectively). Data are expressed as mean ± SEM (*n* = 3, three independent experiments). Unpaired *t*-test was performed. (**C**) FLVCR1a level perturbation affects physical–chemical properties of plasma membrane. Membrane fluidity is expressed as fold increase in normalized Ie/Ii (corresponding to excimer and monomer fluorescence, respectively) fluorescence over their respective controls. Data are expressed as mean ± SEM (*n* = 3, three independent experiments, left panel; *n* = 6, six independent experiments, right panel). Unpaired *t*-test was performed. (**D**) Rescue experiment with iCIC and ALA in FLVCR1a-silenced and -overexpressing Sk-Hep1 cells, respectively. Cholesterol content in plasma membrane is expressed as µmol/mg protein. Data are expressed as mean ± SEM (*n* = 3, three independent experiments). One-way ANOVA test was performed. (**E**) Rescue experiment with iCIC and ALA in FLVCR1a-silenced and -overexpressing Sk-Hep1 cells, respectively. Membrane fluidity is expressed as fold increase in normalized Ie/Ii (corresponding to excimer and monomer fluorescence, respectively) fluorescence over their respective controls. Data are expressed as mean ± SEM (*n* = 3, three independent experiments). One-way ANOVA test was performed. si*FLVCR1a*: FLVCR1a-silenced cells; scramble: respective control; *FLVCR1a^+^*: FLVCR1a-overexpressing cells; empty: respective control; iCIC: mitochondrial citrate carrier inhibitor; ALA: δ-aminolevulinic acid (heme synthesis inhibitor); LSECs: liver sinusoidal endothelial cells; BTECs: breast tumor-derived endothelial cells.

**Figure 4 biomolecules-14-00149-f004:**
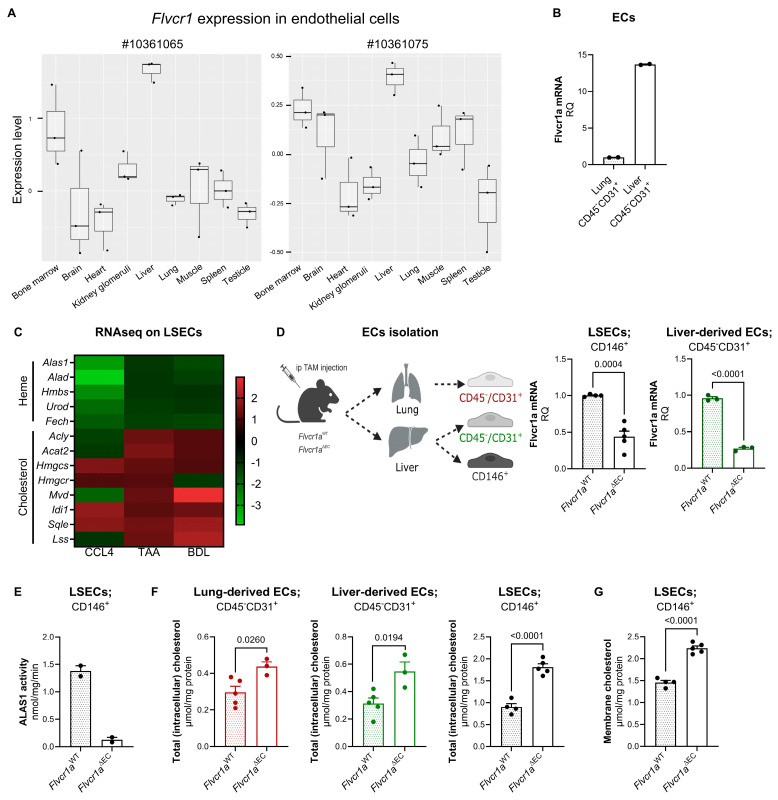
FLVCR1a modulates cellular cholesterol levels in vivo. (**A**) *Flvcr1* expression level in different tissue-specific (microvascular) ECs. Publicly available microarray dataset GSE47067 has been used. *Flvcr1* is represented by two probes on the microarray (#10361065 and #10361075) and expression levels above 0 can be considered to indicate detection. (**B**) mRNA expression levels of *Flvcr1a* in ex vivo lung- and liver-derived ECs. Normalization was achieved using 18S as housekeeping gene. Data are expressed as mean ± SEM of relative quantification using the 2^−∆∆Ct^ method over lung-derived ECs (*n* = 2, each sample is a pool of 3 mice). (**C**) Heat map showing the mRNA expression levels (log_2_ fold change) of heme synthesis- and cholesterol synthesis-related enzymes in LSECs derived from three different rat models of liver cirrhosis (CCL4, TAA, and BDL) versus the healthy ones. Publicly available RNA-seq dataset GSE164878 has been used. CCL4, carbon tetrachloride; TTA, thioacetamide; BDL, bile duct ligation. (**D**) Schematic representation of EC isolation (left) and *Flvcr1a* expression in LSECs (CD146^+^, *n* = 4–5) and ECs (CD45^−^CD31^+^, *n* = 3) isolated from the liver of *Flvcr1a^WT^* and *Flvcr1a^ΔEC^* mice (right). Normalization was achieved using 18S as housekeeping gene. Data are expressed as mean ± SEM of relative quantification using the 2^–∆∆Ct^ method over *Flvcr1a^WT^* derived cells. (**E**) Mitochondrial ALAS activity measured in LSECs (CD146^+^) isolated from the liver of *Flvcr1a^WT^* and *Flvcr1a^ΔEC^* mice. ALAS activity is expressed as nmol/min/mg protein. Data are expressed as mean ± SEM (*n* = 2). (**F**) Total (intracellular) cholesterol content in LSECs (CD146^+^), liver ECs (CD45^−^CD31^+^), and lung ECs (CD45^−^CD31^+^) isolated from *Flvcr1a^WT^* and *Flvcr1a^ΔEC^* mice. Cholesterol content is expressed as µmol/mg protein. Data are expressed as mean ± SEM (*n* = 3–5). Unpaired *t*-test was performed. (**G**) Membrane cholesterol content in LSECs (CD146+) isolated from the liver of *Flvcr1a^WT^* and *Flvcr1a^ΔEC^* mice. Cholesterol content is expressed as µmol/mg protein. Data are expressed as mean ± SEM (*n* = 4–5). Unpaired *t*-test was performed. *Flvcr1a^ΔEC^*: endothelial-specific Flvcr1a-deficient mice; *Flvcr1a^WT^*: respective control mice; LSECs, liver sinusoidal endothelial cells; ECs: endothelial cells.

**Figure 5 biomolecules-14-00149-f005:**
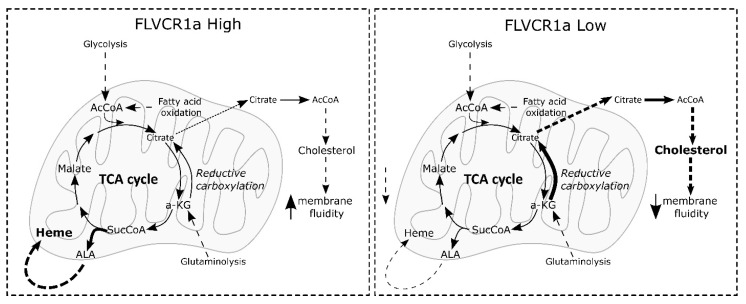
Proposed model showing the inverse relationship between heme synthesis and cholesterol production. Our data demonstrate that FLVCR1a levels and, consequently, heme synthesis (namely, ALAS1-mediated succinyl-CoA consumption) negatively impact cholesterol production, thus ultimately impinging on the physicochemical properties of the plasma membrane. When FLVCR1a expression is elevated, heme synthesis is enhanced, leading to reduced availability of citrate for cholesterol production. Conversely, when FLVCR1a expression is low, the opposite effect occurs. The thickness of the arrows is proportional to the activation of the respective reaction. AcCoA, acetyl-CoA; α-KG, ketoglutarate; ALA, δ-aminolevulinic acid; TCA, tricarboxylic acid; SucCoA, succinyl-CoA.

## Data Availability

The data presented in this study are available in the main text, figures, tables and Appendix A.

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
