# Peer review of "FLVCR1a Controls Cellular Cholesterol Levels through the Regulation of Heme Biosynthesis and Tricarboxylic Acid Cycle Flux in Endothelial Cells"

_biomolecules, 2024, doi:10.3390/biom14020149_

Round 1

Reviewer 1 Report

Comments and Suggestions for Authors

The authors studied the effect of FLVCR1a on metabolic changes in endothelial cells. FLVCR1a is a receptor for a retrovirus, feline leukemia virus C, but it functions as a heme efflux transporter. The author reported in this study that FLVCR1a expression in endothelial cells changes delta-aminolevulinic acid synthesis and cholesterol synthesis. Although the TCA cycle metabolites act as precursors of other metabolic pathways including heme synthesis and cholesterol synthesis, their regulatory processes are not well proven. This study is potentially important that it provides a novel possibility of close link between heme and cholesterol. The results are clear experimental data are clearly shown, however, there are mechanisms for the observations are not fully explained. Te paper is potentially acceptable, but it is nice to improve further discussions.

Specific points.

1.      Line 50. I would like to suggest that the direction of heme transport by FLVCR1a should be clearly described. If the heme is internalized by this transporter, Figure 1 cannot be explained.

2.      Line 71-73. Since FLVCR1a is originally a feline protein, the identity or similarity between the original protein and the human FLVCR1a should be explained in more detail. Also, it should be clearly stated which type of FLVCR1a is utilized in the experiments.

3.      It is little difficult to understand why endothelial cells are used in this study, since heme synthesis and cholesterol synthesis are actively operated in liver cells but not endothelial cells as long as I suppose.

4.      Figure 2H. The data shows glutamate is effectively utilized to produce cholesterol in Sk-Hep1 cells. Fig. 2D illustrates that glutamate can be converted to acetyl-CoA that is the precursor of cholesterol. It seems that the critical reaction in the “reductive carboxylation” that converts alpha-ketoglutaric acid into citrate. However, the process that convert citrate into alpha-ketoglutarate in TCA cycle is an irreversible reaction, so that it cannot be explained by the reverse reaction of TCA cycle. The authors should add a careful explanation regarding this step, “reductive carboxylation”.

Author Response

We are grateful for the reviewer’s critical review of the manuscript and have responded to each comment below. Integrating these suggestions and concerns into the revised manuscript has further improved our manuscript.

Reviewer 1

The authors studied the effect of FLVCR1a on metabolic changes in endothelial cells. FLVCR1a is a receptor for a retrovirus, feline leukemia virus C, but it functions as a heme efflux transporter. The author reported in this study that FLVCR1a expression in endothelial cells changes delta-aminolevulinic acid synthesis and cholesterol synthesis. Although the TCA cycle metabolites act as precursors of other metabolic pathways including heme synthesis and cholesterol synthesis, their regulatory processes are not well proven. This study is potentially important that it provides a novel possibility of close link between heme and cholesterol. The results are clear experimental data are clearly shown, however, there are mechanisms for the observations are not fully explained. Te paper is potentially acceptable, but it is nice to improve further discussions. 

Specific points. 

  1. Line 50. I would like to suggest that the direction of heme transport by FLVCR1a should be clearly described. If the heme is internalized by this transporter, Figure 1 cannot be explained.

In the Introduction, we described FLVCR1a function as a plasma membrane transporter that mediate the transfer of heme from the cytosol to the extracellular milieu.

  1. Line 71-73. Since FLVCR1a is originally a feline protein, the identity or similarity between the original protein and the human FLVCR1a should be explained in more detail. Also, it should be clearly stated which type of FLVCR1a is utilized in the experiments.

We included data regarding the evolutionary conservation of the protein. Furthermore, within the "Materials and Methods" section, we explicitly indicated the utilization of human cells and the overexpression of human FLVCR1a.

  1. It is little difficult to understand why endothelial cells are used in this study, since heme synthesis and cholesterol synthesis are actively operated in liver cells but not endothelial cells as long as I suppose.

Both heme synthesis and cholesterol synthesis are intrinsic and indispensable processes. Heme serves as a co-factor for enzymes/proteins involved in critical functions such as DNA duplication, gene expression, and cell respiration. Conversely, cholesterol plays a crucial role as a fundamental component of cell membranes. As outlined in the introduction, endothelial cells rely on active heme synthesis and a functional FLVCR1a for their optimal function. Furthermore, cholesterol, acting as a constituent and organizer of the plasma membrane, serves as a regulator of endothelial cell mechanical properties. Disruption of cholesterol balance can lead to the impairment of endothelial functions and, ultimately, contribute to the development of diseases. Therefore, endothelial cells provide an ideal model for investigating the impact of the ALAS1-FLVCR1a axis on cholesterol production.

  1. Figure 2H. The data shows glutamate is effectively utilized to produce cholesterol in Sk-Hep1 cells. Fig. 2D illustrates that glutamate can be converted to acetyl-CoA that is the precursor of cholesterol. It seems that the critical reaction in the “reductive carboxylation” that converts alpha-ketoglutaric acid into citrate. However, the process that convert citrate into alpha-ketoglutarate in TCA cycle is an irreversible reaction, so that it cannot be explained by the reverse reaction of TCA cycle. The authors should add a careful explanation regarding this step, “reductive carboxylation”.

Thank you for your comment. We endeavored to elucidate this aspect. α-ketoglutarate, originating from glutamine, undergoes either oxidation or reduction The latter process is catalyzed by mitochondrial and cytosolic isoforms of NADP(+)/NADPH-dependent isocitrate dehydrogenase. We have included this information in the Results section. Additionally, adjustments were made to Figure 2D to more effectively illustrate the two pathways.

Reviewer 2 Report

Comments and Suggestions for Authors

The manuscript by Manco et al., "FLVCR1a Orchestrates Cellular Cholesterol

Levels through the Regulation of Heme Biosynthesis and Tricarboxylic Acid Cycle

Flux in Endothelial Cells" explored the influence of the heme exporter,

FLVCR1, on cholesterol production in endothelial cells. Authors demonstrated that heme export from the cells through FLVCR1 affects tricarboxylic acids cycle (TCA) cataplerosis and supply of citrate for cholesterol biosynthesis.

The authors confirmed their findings in a mouse model and demonstrated the

effect of FLVCR on membrane cholesterol in endothelial cells from the liver and

lung. The manuscript is well-written, and the data presentation is clear. Here are some comments on this work.

Figure 1a and b, bottom panels. The endogenous FLVCR1 in the empty vector control on the bottom panels in Figures 1a and 1 b are not visible since overexpression is

high. Adding a Western image with longer exposure to visualize endogenous

FLVCR1 protein for empty vector control would be advantageous. 

Line 506. The authors isolated endothelial cells from lung and liver tissues using

CD31, CD45, and CD146 magnetic beads. It would be advantageous to confirm the

depletion of FLVCR1 on protein or mRNA levels in cells isolated from

endothelial-specific FLVCR1 knockout animals.

 Line 510. And figure 4E. "As expected, ALAS ac-510 activity was reduced in

endothelial cells isolated from Flvcr1aΔEC mice compared with those from Flvcr1aWT mice (Figure 4E)".

There are discrepancies between the text (line 510) "endothelial cells" and

labeling in Figure 4E (LSEC). It would be more apparent to indicate what cells

were used for analysis CD146+ or CD45-/CD31+ in the figure and not only the

figure legend.

Please provide catalog number CD146 microbeads in section 2.6. EC Isolation, Materials and Methods.

Figure 5. This figure is a good summary of the work. The arrow showing the inhibition

effect of SucCoA on aKG is very light and not evident. It would be helpful to

make this arrow more visible.

 The explanation for the TCA abbreviation is provided only in Figure 5. Please add the tricarboxylic acid (TCA) abbreviation at the beginning of the paper or

in the list of abbreviations.

Author Response

We are grateful for the reviewer’s critical review of the manuscript and have responded to each comment below. Integrating these suggestions and concerns into the revised manuscript has further improved our manuscript.

Reveiewer 2

The manuscript by Manco et al., "FLVCR1a Orchestrates Cellular Cholesterol Levels through the Regulation of Heme Biosynthesis and Tricarboxylic Acid Cycle Flux in Endothelial Cells" explored the influence of the heme exporter, FLVCR1, on cholesterol production in endothelial cells. Authors demonstrated that heme export from the cells through FLVCR1 affects tricarboxylic acids cycle (TCA) cataplerosis and supply of citrate for cholesterol biosynthesis. The authors confirmed their findings in a mouse model and demonstrated the effect of FLVCR on membrane cholesterol in endothelial cells from the liver and lung. The manuscript is well-written, and the data presentation is clear. Here are some comments on this work.

Figure 1a and b, bottom panels. The endogenous FLVCR1 in the empty vector control on the bottom panels in Figures 1a and 1 b are not visible since overexpression is high. Adding a Western image with longer exposure to visualize endogenous FLVCR1 protein for empty vector control would be advantageous. 

We have shown the images with longer exposure in the supplement “uncropped gels”.

Line 506. The authors isolated endothelial cells from lung and liver tissues using CD31, CD45, and CD146 magnetic beads. It would be advantageous to confirm the depletion of FLVCR1 on protein or mRNA levels in cells isolated from endothelial-specific FLVCR1 knockout animals.

We added these data in Figure 4D.

Line 510. And figure 4E. "As expected, ALAS ac-510 activity was reduced in endothelial cells isolated from Flvcr1aΔEC mice compared with those from Flvcr1aWT mice (Figure 4E)". There are discrepancies between the text (line 510) "endothelial cells" and labeling in Figure 4E (LSEC). It would be more apparent to indicate what cells were used for analysis CD146+ or CD45-/CD31+ in the figure and not only the figure legend.

Thank you for this comment. We modified the figure accordingly.

Please provide catalog number CD146 microbeads in section 2.6. EC Isolation, Materials and Methods.

We added this information.

Figure 5. This figure is a good summary of the work. The arrow showing the inhibition effect of SucCoA on aKG is very light and not evident. It would be helpful to make this arrow more visible.

Thank you for your comment. We have addressed this by adding clarification in the figure legend, specifying that the arrow thickness is directly proportional to the activation levels of both heme synthesis and cholesterol synthesis reactions.

The explanation for the TCA abbreviation is provided only in Figure 5. Please add the tricarboxylic acid (TCA) abbreviation at the beginning of the paper or in the list of abbreviations.

We added the abbreviation in the Introduction.
